# Machine learning optimization of candidate antibody yields highly diverse sub-nanomolar affinity antibody libraries

Lin Li [1] ✉, Esther Gupta[1], John Spaeth[1], Leslie Shing [1], Rafael Jaimes [1], Emily Engelhart[2], Randolph Lopez[2], Rajmonda S. Caceres[1,6], Tristan Bepler [3,4,6] & Matthew E. Walsh[1,5,6]

Therapeutic antibodies are an important and rapidly growing drug modality. However, the design and discovery of early-stage antibody therapeutics remain a time and cost-intensive endeavor. Here we present an end-to-end Bayesian, language model-based method for designing large and diverse libraries of high-affinity single-chain variable fragments (scFvs) that are then empirically measured. In a head-to-head comparison with a directed evolution approach, we show that the best scFv generated from our method represents a 28.7-fold improvement in binding over the best scFv from the directed evolution. Additionally, 99% of designed scFvs in our most successful library are improvements over the initial candidate scFv. By comparing a library's predicted success to actual measurements, we demonstrate our method's ability to explore tradeoffs between library success and diversity. Results of our work highlight the significant impact machine learning models can have on scFv development. We expect our method to be broadly applicable and provide value to other protein engineering tasks.

Therapeutic antibodies are an important and rapidly growing drug modality. Because the vast search space of antibody sequences renders exhaustive evaluation of the entire antibody space infeasible, screening relatively small numbers of antibodies from synthetic generation, animal immunizations or human donors are used to identify candidate antibodies. The screened library represents a small portion of the overall search space, and the resultant candidate antibodies are often weak binders or suffer from developability issues. Optimization of these candidates is needed to improve binding and other development characteristics.

Due to the combinatorial scaling of sequence space, step-wise, iterative approaches are often used to optimize antibody binding against target molecules[1,2], but are time consuming and effort is wasted interrogating nonfunctional antibodies. Improved binders may

need to be further altered to improve other properties, such as hydrophobicity[3,4], but such alterations can negatively influence the previously optimized binding, resulting in additional measurement and engineering cycles. This process of identifying the final antibody routinely takes about 12-months to complete[2]. The ability to efficiently engineer antibodies with favorable binding and high diversity earlier in the development process would reduce the impact of unfavorable antibody characteristics that are often identified later in the process, improve the developability potential and reduce the time required in early drug development.

While computational methods can guide the search of biologically relevant antibodies, most de novo approaches require target structures or antibody-epitope complex structures to be known[5–7]. Machine learning (ML) approaches can be used to effectively represent

[1]Massachusetts Institute of Technology Lincoln Laboratory, Lexington, MA, USA. [2]A-Alpha Bio, Inc., Seattle, WA, USA. [3]Research Laboratory of Electronics, Massachusetts Institute of Technology, Cambridge, MA, USA. [4]Present address: Simons Electron Microscopy Center, New York Structural Biology Center, New York, NY, USA. [5]Present address: Johns Hopkins Bloomberg School of Public Health, Baltimore, MD, USA. [6]These authors contributed equally: Rajmonda S. Caceres, Tristan Bepler, Matthew E. Walsh. ✉e-mail: Lin.Li@LL.MIT.EDU

biological data and rapidly explore their vast design spaces in silico. Such approaches can uncover complex and flexible features from high-dimensional data[8–13] and have shown great promise in many application areas, including protein structure prediction[14], and drug discovery and design[15–21]. Existing ML-driven antibody optimization has shown promising results in designing antibodies with improved binding characteristics against a target and that antibody binding can be learned from only sequence data and without the need for the target's structure[15]. A more recent work has presented an ML-driven antibody optimization approach that achieves broader neutralizing activity against diverse SARS-CoV-2 variants by learning the mutational effect on protein-protein interactions from protein complex structures[20]. Other works have investigated general purpose pre-trained generative language models for designing antibody libraries that display good physical properties[18,19], but these methods are not target-specific and only offer modest improvements over conventional libraries that are, often, already based on natural antibody repertoires. Finally, none of the existing work allows the evaluation of designed antibody libraries prior to experimentation, a critical feature that allows for accelerated design cycles.

In this work, we develop an end-to-end ML-driven single-chain variable fragment (scFv) design framework that uniquely combines state-of-art language models, Bayesian optimization and high-throughput experimentation (Fig. 1). Because we synthesize explicitly defined oligo pools of 300 bp, our method allows the design of the entire scFv chain (heavy or light). Furthermore, it does not assume candidate scFvs strongly bind to the target, and relies on sequence data without the need for sequence alignments or knowledge of the target antigen structure, allowing the method to be applicable to early-stage antibody development for any target antigen. We demonstrate our end-to-end framework can rapidly and cost-effectively lead to the design of diverse target-specific scFv libraries with therapeutically relevant binding affinities. At a meaningful scale (~10^4 sequences), and in a head-to-head comparison with the directed evolution approach, we show that our ML-based approach produces significantly stronger binders. More remarkably, our ML-designed scFv libraries are highly diverse, demonstrating the ability of our approach to efficiently extrapolate and discover mutationally distant, high affinity scFvs. Lastly, we show how our method can provide general insights to the engineering process. We can evaluate the performance of an scFv

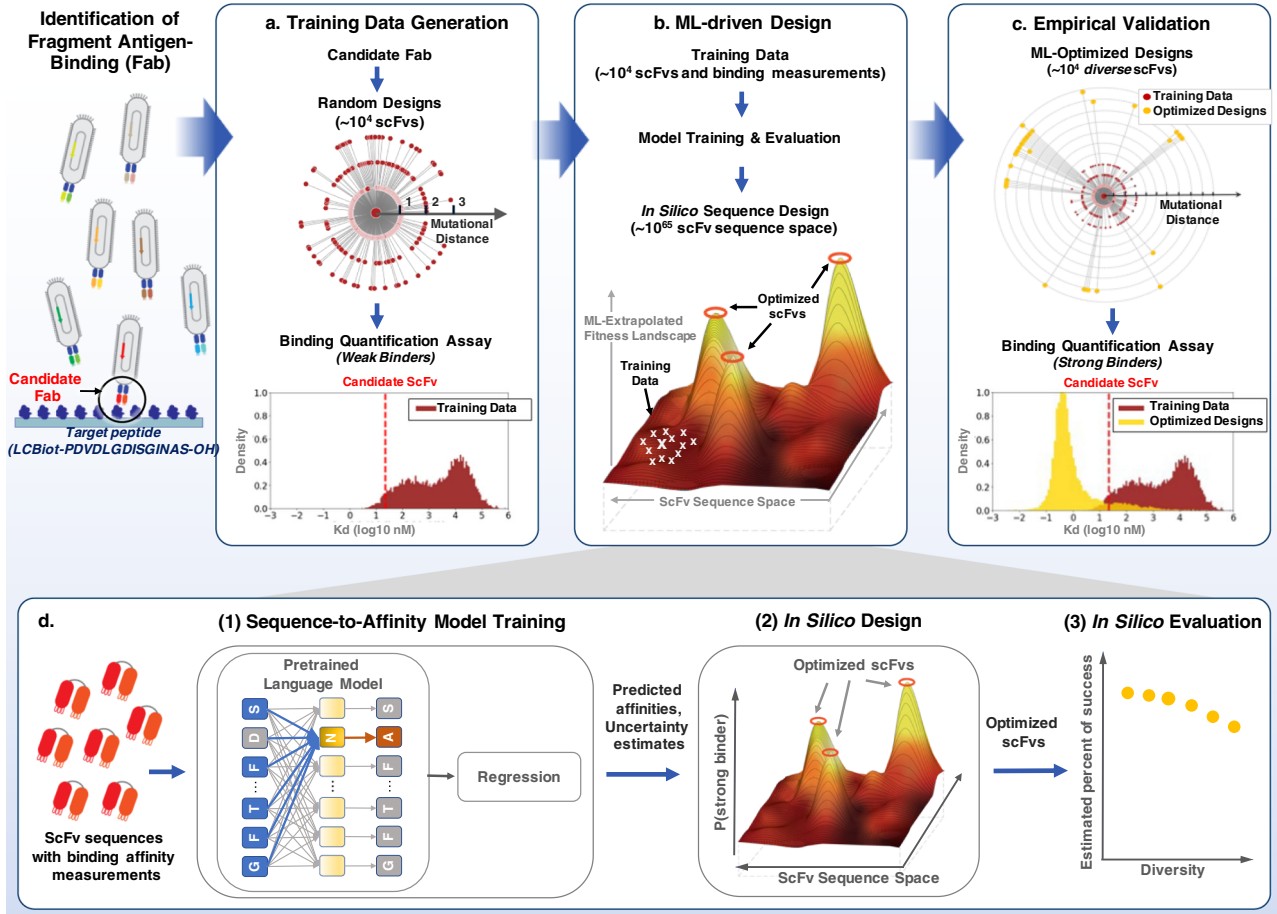

**Fig. 1 | Illustration of the end-to-end ML-driven scFv design process.** The end-to-end process consists of three components: training data generation, ML-driven design to generate scFv libraries and empirical validation of designed libraries, providing a pool of potential scFv candidates for further development. **a** The training data is generated via random mutations of the candidate scFv along the entire CDR region, followed by high-throughput binding quantification to the selected target. The circular plot is a conceptual description of scFv sequences used during training; each circle represents the sequence space with its associated mutation number from the candidate scFv (the center of the diagram). Sequences are placed uniformly at random along each mutation circle because the training data are generated uniformly at random from the candidate sequence. **b** This training data combined with publicly available protein sequences is used to train, refine and evaluate ML models that drive the in silico sequence design process. **c** The designed libraries are experimentally validated, providing thousands of potential antibody candidates for development. ML-driven designs produce highly diverse scFvs (sequences as far as 23 mutations away), with strong on-target binding (the best design is 28.7-fold better than the directed evolution approach), and high success rate (as high as 99%). **d** Detailed ML-driven design process: (1) supervised fine-tuning of pretrained language models on the training data to predict binding affinities with uncertainty quantification; (2) in silico scFv design via Bayesian optimization over ML-extrapolated fitness landscape; (3) in silico scFv library evaluation.

library in silico, explore the affinity-diversity tradeoff prior to experimental testing, weigh the choice of optimizing complementarity-determining regions (CDRs) jointly or individually, and combine our method with other software tools to explore other desired development properties, such as hydrophobicity and isoelectric point, of scFvs in designed libraries. Our results highlight the impact ML models can have on early-stage scFv development. Through coordinated data generation, ML model development, training and optimization, we are able to start with only a target protein sequence and a candidate antigen-binding fragment (Fab) that weakly binds to the target, and after a single round of optimization, generate large, diverse libraries of high-affinity scFvs against the target.

## Results

### Development of an end-to-end, target-specific scFv optimization process

We hypothesized that by integrating target-specific binding affinities with information from millions of natural protein sequences in a probabilistic machine learning framework, we could rapidly engineer scFvs that are significantly stronger binders than what typical directed evolution approaches would produce. To engineer a given candidate scFv (the variable fragment of a Fab) against the target molecule, we developed a five-step process that uniquely combines state-of-art language models, Bayesian optimization and high-throughput experimentation to generate high-affinity scFv libraries (Fig. 1 and Methods section):

1. High-throughput binding quantification of random mutants of the candidate scFv to the target, to create supervised training data (Fig. 1a).
2. Unsupervised pre-training of language models[22,23] on large numbers of protein sequences to distill biologically relevant information and represent scFv sequences (Fig. 1b, d).
3. Supervised fine-tuning of pretrained language models on the training data to predict binding affinities with uncertainty quantification (Fig. 1b, d).
4. Construction of a Bayesian-based scFv fitness landscape extrapolated from the trained sequence-to-affinity model, followed by in silico scFv design via Bayesian optimization and in silico design validation (Fig. 1b, d).
5. Experimental validation of top scFv sequences that are in silico predicted to have strong binding affinities for the target (Fig. 1c).

We generated our supervised training data using an engineered yeast mating assay. The target peptide is a conserved sequence found in the HR2 region of coronavirus spike proteins and to which neutralizing antibodies were previously identified[24]. A phage display campaign with a phage library containing naïve human Fabs was used to identify candidate scFv sequences (Ab-14, Ab-91, and Ab-95) that bind weakly to the target (Supplementary Table 1). All heavy and light chain sequences in the data were designed by performing random $k = 1, 2, 3$ mutations within either the heavy chain or light chain CDRs of three candidate scFvs (Supplementary Table 2). We have separately published this dataset in its entirety to support its reuse[25,26]. In this work, we sought to optimize Ab-14 and therefore used only the Ab-14 measurements (26,453 heavy chain, 26,223 light chain) as supervised training data for the sequence-to-affinity prediction. The binding measurements are provided on a log-scale, with lower values indicating stronger binding; see Supplementary Fig. 1 for the distribution of binding measurements.

We pre-trained four BERT masked language models, i.e., a protein language model, an antibody heavy chain model, an antibody light chain model and a paired heavy-light chain model. The protein language model was trained on the Pfam data[27], and antibody-specific language models were trained on human naïve antibodies from the Observed Antibody Space (OAS) database[28] (Supplementary Table 3).

To train sequence-to-affinity models, we investigated two approaches to predict affinities with uncertainty quantification: an ensemble method and Gaussian Process (GP)[29]. Both approaches use learned knowledge from pre-trained language models and provide meaningful sequence-to-affinity models from which one can design high-affinity scFv libraries. We trained separate sequence-to-affinity models for Ab-14-H heavy-chain variants and Ab-14-L light-chain variants using the corresponding training data. We observed strong positive correlation between predicted and experimentally measured binding affinities on the hold-out test data (Supplementary Fig. 2).

To generate high-affinity scFv libraries, a Bayesian-based fitness landscape was constructed to map the entire scFv sequence to a posterior probability, i.e., the probability that the estimated binding affinity is better than the candidate scFv Ab-14. This is in contrast to the fitness landscape that goes directly from sequence to estimated binding affinity. To perform optimization to maximize the posterior probability, the choice of sampling algorithm is critical in determining the library diversity. Three strategies were used: hill climb (HC), genetic algorithm (GA) and Gibbs sampling. HC is a greedy algorithm that performs a local search and only finds local maximums. GA is an evolutionary-based algorithm that is more robust in exploiting the sequence space further away from the initial sequence. Gibbs sampling takes sequential actions in a manner that balances exploitation and exploration and can generate sequences with high diversity.

We applied our sampling approaches to generate heavy chain and light chain variant scFvs that optimize Ab-14. We also used a Position-Specific Score Matrix (PSSM)-based method representative of traditional directed evolution approaches to generate a control sequence set. The generated sequences from each method are rank-ordered based on the posterior probability and top sequences are selected. This resulted in seven scFv libraries per chain: three libraries from optimizing the ensemble-based fitness function (namely, En-HC, En-GA and En-Gibbs), three libraries from optimizing the GP-based fitness function (namely, GP-HC, GP-GA, GP-Gibbs), and one PSSM library (Supplementary Figs. 3 and 4). As a sanity check, we also generated scFv mutants with an average of $k = 2$ random mutations from the 10 strongest binders of the supervised training data. All sequences were synthesized and experimentally tested using the same high-throughput yeast display method as for the training data generation; Supplementary Tables 4 and 5 provide the exact number of sequences from each library.

We compared the empirical binding distribution of the training data with the PSSM library and ML-designed sequences (Supplementary Fig. 1). ML designs are significantly stronger binders than the training data. Notably, more than 25% of ensemble-based Ab-14-H variant designs have stronger measured binding affinities than the strongest measured binder in the training data, whereas only 0.9% of PSSM-based Ab-14-H variant designs outperform the strongest measured binder in the training data.

### ML-generated ScFv libraries outperform conventional directed evolution

We assessed the quality of each ML-derived scFv library by comparing the binding strength of the best design and the percent of success to the PSSM-generated library. We define the percent of success as the percent of scFvs that have a better empirical binding score than the initial candidate scFv, Ab-14. We chose PSSM libraries as comparators because they better reflect the traditional optimization process and are generally better than random mutation libraries (Supplementary Fig. 5). Table 1 contains characterization of the best binding scFv from each library. Sequences of these scFvs can be found in the Supplementary Tables 6 and 7. The best scFvs from ML-optimized libraries are significantly stronger binders than those from the PSSM library, and generally have more mutations. The strongest binding heavy-chain design is from the En-Gen library and binds 28.7-fold stronger than the

**Table 1 | Characterization of the top scFv from each library**

| Library | Best Ab-14-H variant design | | | Best Ab-14-L variant design | | |
|---|---|---|---|---|---|---|
| | Predicted affinity (pM) | Mutational distance to Ab-14 | Fold improvement over PSSM | Predicted affinity (pM) | Mutational distance to Ab-14 | Fold improvement over PSSM |
| PSSM | 109.602 | 4 | 1.0 | 113.053 | 3 | 1.0 |
| GP-HC | 52.179 | 3 | 2.1 | 57.944 | 3 | 2.0 |
| GP-GA | 20.483 | 4 | 5.4 | 16.454 | 3 | 6.9 |
| GP-Gibbs | 15.541 | 4 | 7.1 | 98.980 | 9 | 1.1 |
| En-HC | 3.817 | 7 | **28.7** | 156.090 | 11 | 0.7 |
| En-GA | 3.923 | 10 | 27.9 | 30.400 | 17 | 3.7 |
| En-Gibbs | 38.126 | 15 | 2.9 | 14.608 | 23 | **7.7** |

Bold values represent the highest fold improvement over the best PSSM sequences.

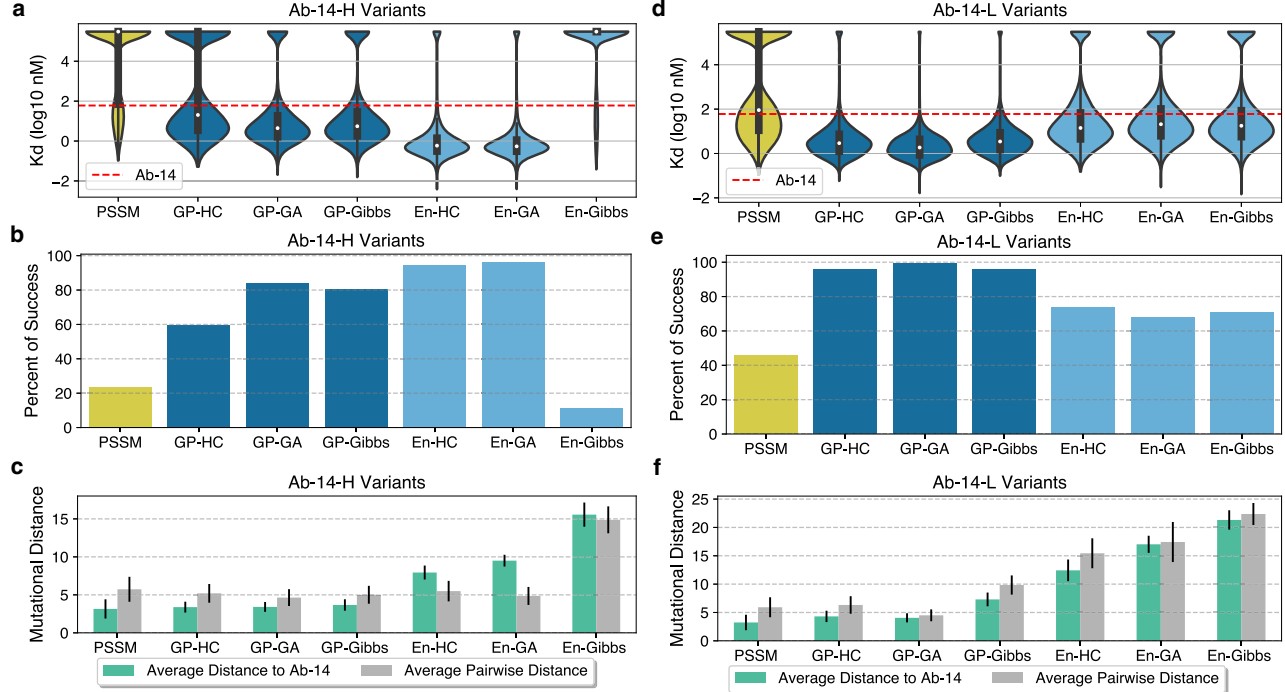

**Fig. 2 | ML-optimized scFv libraries outperform the PSSM directed evolution approach with high percentage of success and high diversity.** For sequences with at least 3 (out of 6) empirical binding affinities, averaged values are used as the ground-truth. The rest of the sequences (with less than 3 empirical measurements) are considered as un-successful designs. All evaluations are performed over $n = 6510, 5152, 5313, 5284, 5344, 5310, 4879$ Ab-14-H variant designs and $n = 8188, 5965, 5989, 5987, 5962, 5960, 5950$ Ab-14-L variant designs generated by PSSM, GP-HC, GP-GA, GP-Gibbs, En-HC, En-GA and En-Gibbs, respectively (see Supplementary Tables 4 and 5), where GP and En denote Gaussian Process and Ensemble models, and HC, GA and Gibbs denote hill climb, genetic and Gibbs sampling algorithms, respectively. **a** The violin plot is used to depict summary statistics and empirically measured affinity distribution of Ab-14-H heavy chain designs (center: median; limits: 1st and 3rd quartile; whiskers: +/− 1.5 IQR). Affinities of unsuccessful sequences are set to be 5.48 (the largest assay value of all Ab-14-H variants). **b** Percent of sequences that have stronger empirical binding affinity than the candidate antibody for all the Ab-14-H variant libraries. **c** Diversity comparison for all Ab-14-H variant libraries. Data are presented as mean values and +/- standard deviation to show mutational variability of designed sequences from the initial candidate scFv. **d** The violin plot is used to depict summary statistics and empirically measured affinity distribution of Ab-14-L light chain designs (center: median; limits: 1st and 3rd quartile; whiskers: +/− 1.5 IQR). Affinities of unsuccessful sequences are set to be 5.53 (the largest assay value of all Ab-14-L variants). **e** Percent of success for all the Ab-14-L variant libraries. **f** Diversity comparison for all the Ab-14-L variant libraries. Data are presented as mean values and +/- standard deviation. Source data are provided as a Source Data file.

strongest scFv in the PSSM library. The best light-chain design is in the En-Gibbs library achieving a 7.7-fold improvement over the best scFv from the PSSM library. Note that the best heavy-chain scFv binds much stronger to the target than the best light-chain scFv. To investigate further, we rank-ordered all designed scFvs across different libraries by the empirically-measured binding affinity and observed that heavy-chain designs are generally stronger binders than light-chain designs (Supplementary Fig. 6).

Figure 2 shows the performance and diversity of designed libraries. For Ab-14-H heavy chain designs, with the exception of sequences

in the En-Gibbs library, all ML-optimized libraries outperform the PSSM library in terms of median binding affinity (Fig. 2a), and are significantly more successful than the 23.8% success of the PSSM library (Fig. 2b). The En-HC (94.3%) and En-GA (96%) libraries are particularly successful and outperform all GP-generated Ab-14H variant libraries (59.4–84.2%). For the Ab-14-L light chain designs, all ML-optimized libraries outperform the PSSM library in both median binding (Fig. 2d) and percent of success whereas the PSSM library is 45.6% successful (Fig. 2e). The percent of success of GP-based libraries (95.7–99%) further outperforms all ensemble-based libraries (67.9–73.5%).

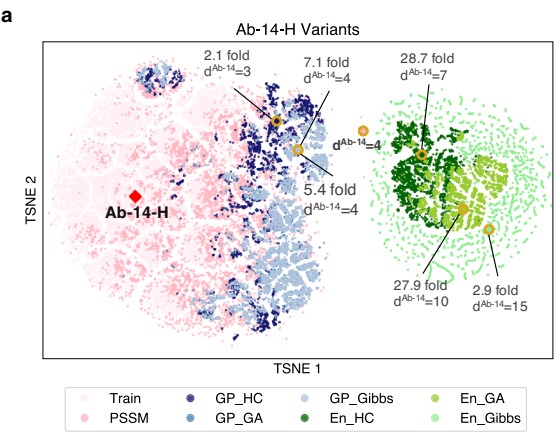

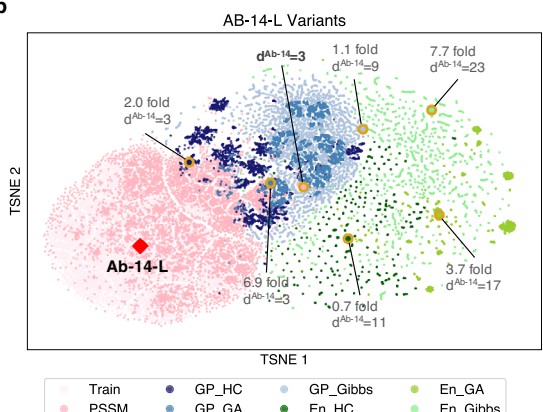

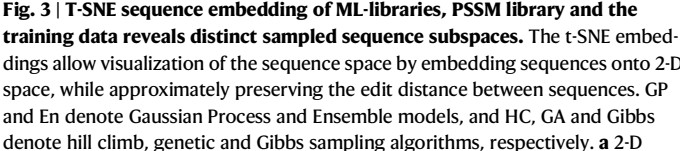

**Fig. 3 | T-SNE sequence embedding of ML-libraries, PSSM library and the training data reveals distinct sampled sequence subspaces.** The t-SNE embeddings allow visualization of the sequence space by embedding sequences onto 2-D space, while approximately preserving the edit distance between sequences. GP and En denote Gaussian Process and Ensemble models, and HC, GA and Gibbs denote hill climb, genetic and Gibbs sampling algorithms, respectively. **a** 2-D embeddings of Ab-14-H variants. **b** 2-D embeddings of Ab-14-L variants. The initial candidate sequence Ab-14 is marked with a diamond marker. The best scFv variants from each library are marked in circles. The best ML-generated scFvs are labeled with fold improvement over the best PSSM scFv and the mutational distance from the candidate Ab-14 scFv. The best PSSM scFv is labeled with mutational distance only. Source data are provided as a Source Data file.

## ML-generated libraries can be highly diverse

We measured the library diversity using two mutational distance metrics: $d_{avg}^{Ab-14}$ (the average distance to the initial Ab-14), and $d_{pw}$ (the average pairwise distance). The former $d_{avg}^{Ab-14}$ indicates how far the designs are from the training data and the latter $d_{pw}$ measures the intra-library diversity. For Ab-14-H variant designs, all ML-optimized libraries have higher $d_{avg}^{Ab-14}$ than the PSSM library (with $d_{avg}^{Ab-14} = 3.1$). The ensemble-based libraries also have significantly higher $d_{avg}^{Ab-14}$ (7.9–15.6) than the GP-based libraries (3.4–3.7), indicating that the methods are able to extrapolate and design sequences that are far beyond the training data (Fig. 2c). In particular, sequences in the En-Gibbs library are on average 15.6 distance away from Ab-14-H and 14.9 distance away from each other (Fig. 2c). However, this increase in mutational distance comes at the cost of reduced affinity, suggesting that there is eventually a tradeoff between the two.

For Ab-14-L variant designs, all ML-optimized libraries are significantly further away from Ab-14-L than the PSSM library, with $d_{avg}^{Ab-14} = 3.2$ for the PSSM library, $d_{avg}^{Ab-14}$ ranging from 4.3 to 7.4 for GP-based libraries and $d_{avg}^{Ab-14}$ ranging from 12.4 to 21.3 for ensemble-based libraries (Fig. 2f). With the exception of GP-GA ($d_{pw} = 4.5$), all ML-optimized libraries have higher $d_{pw}$ (ranging from 6.3 to 22.4) than the PSSM library ($d_{pw} = 5.9$). In particular, the En-Gibbs light-chain library consists of sequences that are on average 21.3 distance away from Ab-14-L and 22.4 distance away from each other (Fig. 2f).

Figure 3 shows the 2-D embeddings of all scFv libraries and the training data. We observed a similar trend for both light- and heavy-chain designs, that is, the PSSM library is the closest to the training data while the ensemble-based libraries are the farthest away from the training data. More interestingly, all optimization-based libraries occupy a distinct subspace from the training data and PSSM library, highlighting the extrapolating power of the various optimization approaches that we applied. Ensemble-based libraries are highly divergent and also group distinctly from the other libraries; both the best heavy- and light-chain designs were discovered via optimizing the ensemble-extrapolated fitness function, underlining the value of exploring further away from the initial candidate sequence.

## Model performance and sampling diversity are key factors in generating a quality library

To understand key factors that determine the quality of a generated library, we evaluated the performance of the two sequence-to-affinity models, using held-out test data and empirical binding measurements

of our designed sequences (Fig. 4). We compared the Spearman correlation and the mean absolute error (MAE) of model predictions and measured values. We observed that the ensemble sequence-to-affinity model does better at predicting affinity than the GP model. When evaluated on the held-out test data, Spearman correlation scores of both heavy- and light-chain ensemble models are slightly higher (heavy-chain model: 0.51; light-chain model: 0.69) than the respective GP models; see Fig. 4a. When evaluated on designed Ab-14-L variants, the light-chain ensemble model is also slightly better. The most notable difference is when evaluating on designed Ab-14-H variants, where the heavy-chain ensemble model has a Spearman correlation of 0.69 but the heavy-chain GP model performs significantly worse (−0.42). This is primarily due to the prediction limit of the GP model on sequences that are far beyond the training data. When evaluated the MAE of our prediction models with respect to the mutational distance on designed sequences, we observed a sharp increase in MAE on sequences with six or more mutations away from Ab-14-H for the heavy-chain GP model, and on sequences with ten or more mutations away from Ab-14-L for the light-chain GP model (Fig. 4b). Ensemble models exhibit no notable increase in MAE as the mutational distance increases, indicating that the ensemble approach is more generalizable to higher-order mutants than the GP model. Nevertheless, GP-based libraries, when compared to the PSSM library, are significantly more successful while having comparable sequence diversity (Fig. 2).

While ML-guided exploration of sequence space allows for identification of more scFvs with optimized binding, we postulate that if this set comes from diverse sequence space, it will also have diverse development properties thus limiting the chance of correlated downstream development failure. We observed that a good prediction model is necessary but not sufficient to generate a diverse library with high affinity. Equally important to the prediction model is the choice of sampling algorithm. When using the ensemble-extrapolated fitness landscape to engineer 14-Ab-H, hill climb and genetic algorithms found scFvs with significant (28.7 and 27.9-fold, respectively) increases in binding over the best PSSM-sampled scFv (Table 1), and both methods were highly successful (94.3% and 96% success, respectively); see Fig. 2a, b. However, when combined with the Gibbs sampling algorithm, the best scFv sampled was only 2.9-fold better (Table 1), and the library was generally unsuccessful (Fig. 2a, b). With the diversity metrics of the En-Gibbs-generated sequences almost double that of the En-HC and En-GA libraries, it indicates that the significant increase in diversity of the En-Gibbs library has a detrimental effect on library affinity due to the

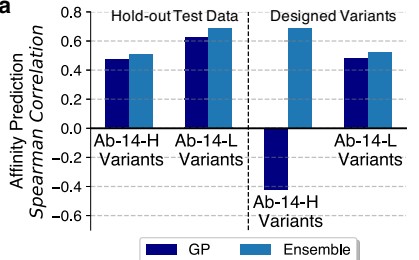
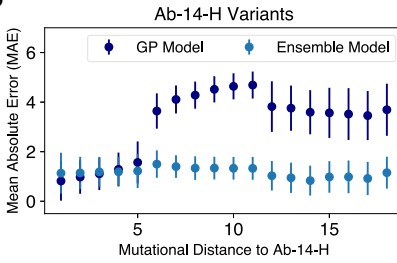
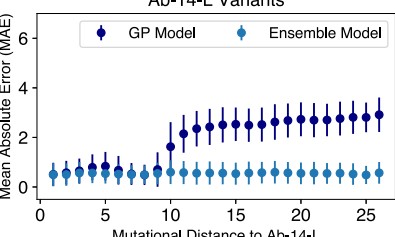

**Fig. 4 | Sequence-to-affinity model evaluation.** All evaluations were performed on sequences with at least 3 (out of 6) empirical binding affinities and the averaged values are used as the ground-truth. GP denotes Gaussian Process. **a** Regression performance on hold-out test data and on the designed libraries; the ensemble model is more predictive than the GP model on both datasets. **b** Performance of GP and ensemble models with respect to mutational distance from Ab-14. Data are presented in mean absolute error (MAE) and +/-SEM. The sample sizes of Ab-14-H variants for mutational distances ranging from 1 to 18 are $n$ = 93, 1337, 4485, 5009, 834, 296, 1316, 2400, 2855, 2304, 418, 53, 131, 148, 152, 124, 71, 33, respectively. The sample sizes of Ab-14-L variants for mutational distances ranging from 1 to 26 are $n$ = 258, 1784, 3696, 6287, 4168, 2097, 2037, 1748, 932, 675, 1042, 1095, 1109, 888, 933, 1447, 1632, 1090, 1025, 1063, 1317, 1168, 741, 341, 86, 18, respectively. Ensemble models are more robust at extrapolating mutationally distant scFvs while the GP models do not predict well on sequences that are mutationally far away from Ab-14. Note that the error bar of the heavy-chain ensemble model shows a non-trivial increase on sequences that are twelve or more mutations away from Ab-14, suggesting that the model's predictability decreases with increase in mutational distance. Source data are provided as a Source Data file.

eventual limit of the model predictability on sequences that are deemed too far from the training data (Fig. 2c). Interestingly, when engineering the light chain (14-Ab-L), the En-Gibbs combination found the strongest binder (7.7-fold improvement over PSSM) with a striking 23 mutations from the Ab-14-L sequence (Table 1). For the ensemble-based libraries, as the library diversity increased, so too did the binding strength of its top scFv (Fig. 2f and Table 1). En-HC, the least diverse ensemble-generated 14-Ab-L library, was the only library that failed to contain an scFv outperforming the top PSSM-generated scFv (Table 1). In this instance, the increased library diversity is beneficial, suggesting the value in exploring away from the initial candidate sequence. Hence, to avoid unsuccessful library designs while still being able to explore sufficiently high orders of mutants, it is important to control the diversity of sampled sequences via parameter tuning of the sampling algorithm and have the ability to explore the tradeoff between performance and diversity in silico prior to experimental testing.

**Bayesian-based approach provides insights prior to experimental testing**

We defined an in silico performance metric that quantifies the binding performance of a library prior to experimental testing. With the Bayesian approach, the fitness score is the posterior probability of a sequence in the library having a stronger binding affinity than the candidate scFv Ab-14. We average the individual fitness scores of the full library to come up with our metric - an estimate of the probability of success (i.e., the estimated percent of sequences having a better binding performance than the threshold value). We first evaluated the utility of the metric on the hold-out test data from the training scFv library as we vary the threshold value that defines strong binders and show the estimated percent of success matches well to the actual percent of success (Supplementary Fig. 7).

We applied the metric (estimated percent of success) to the designed libraries and ranked them. We compared the library ranking based on the estimated and measured percent of success (Supplementary Table 8). For PSSM and ensemble-based libraries, the predicted rankings match well to the actual rankings with a rank correlation of 0.8. For ranking PSSM and GP-based libraries, the metric predicts all rankings correctly for Ab-14-H variant libraries and a rank correlation of 0.8 for Ab-14-L variant libraries. Moreover, we observed that the estimated percent of success captures well the relative performance of designed libraries for both heavy- and light-chain designs (Supplementary Figs. 8 and 9).

We then sought to extend the application of the in silico metric to comparing the choice of optimizing one CDR to optimizing all three simultaneously. For this comparison, designs were generated using the genetic algorithm sampling over the ensemble-extrapolated fitness landscape. We observed that designing all heavy-chain CDRs leads to sequences with higher estimated percent of success than when designing individual CDRs (Supplementary Fig. 10).

Based on these findings, we demonstrate that the performance metric can be used to understand design choices and explore tradeoffs between performance and diversity, and in the future to inform library selection and parameter tuning prior to experimental testing.

## Discussion

We demonstrate, in a head-to-head comparison with a conventional directed evolution strategy, scFvs designed with our ML approach are significantly stronger binders, especially at high levels of diversity, where, remarkably, our models are able to accurately predict binding affinity for extremely high order mutants. Notably, after a single round of design-build-test cycle, we are able to generate a heavy-chain scFv that binds 28.7-fold stronger than the strongest scFv in the PSSM library (Table 1). Most of ML-designed scFvs are improvements over the candidate scFv Ab-14; more than 90% of the empirically evaluated En-GA and En-HC heavy-chain scFvs are successful as compared to less than 20% of success for the PSSM library (Fig. 2). Moreover, the ensemble-based method is able to explore a significantly larger sequence space; the average mutational distance of heavy-chain ensemble libraries ranges from 7.9 to 15.6 as compared to 3.17 of the PSSM library (Figs. 2 and 3). The conventional approach could eventually find a binder as strong as the best binder found with our ML approach. However, this is not guaranteed. At the minimum, it would require at least one additional design-build-test cycle. A single cycle of our process is on the order of a few months, making the conventional approach significantly less time and cost efficient in the best case. The conventional method is unlikely to ever reach equivalent percent of success and diversity metrics. Future work with the intent of quantifying the cost and time saving of integrating machine learning methods in therapeutic development to discover better therapeutics faster would support broader adoption of such methods and should be pursued.

Libraries generated through our method also have diverse biophysical properties as computed using BioPython[30] (Supplementary Fig. 11). This allows for the selection of multiple preclinical candidates, uncorrelated in their downstream failure modes, such that if one fails, the entire pipeline is not likely to fail for the same reason. In the future, biophysical properties that are known to be associated with developability or physiochemical properties can be included in the library

design criteria, such as designing strong binders within specific iso-electric point or hydrophobicity ranges. We also believe that our framework is applicable to any task aiming to maximize or minimize a characteristic of an scFv, such as minimizing off-target binding or maximizing neutralization. Pending data availability, we see ML-based multi-objective scFv optimization as an approachable task and viable option for streamlining scFv development.

We separately explored our model performance as a function of the amount of training data and demonstrated additional data, expectedly, results in improved performance[31]. However, after about 7000 measurements, additional measurements result in less significant performance increases. For this work, we trained our supervised sequence-to-affinity models on all measurements that were available to us, but future engineering attempts may optimize use of financial resources by increasing the number of cycles or number of candidate scFvs to be optimized while reducing the number of measurements per cycle per candidate scFv. Because of cost limitations associated with DNA synthesis, we chose to generate our training data by introducing $k = 1, 2,$ or 3 random mutations, but our models are able to successfully extrapolate much further than that. Future work would benefit from an improved understanding of the way in which training data is generated, if there is dependence on the choice of model, and if performing multiple measurement cycles impacts the choice. We also compared the performance of sequence-to-affinity models with and without pretrained language models[28]. We found that models fine-tuned from pretrained language models outperform models without a pretrained language model, as well as simpler encoding methods like PSSM-based encoder. The Pfam pretrained language model performs better than OAS pretrained language models. We postulate that learning from the more diverse protein sequences captures higher level biological principles that can be transferred and refined more effectively to antibody specific tasks such as affinity prediction.

Recently, other works have presented general purpose pre-trained generative language models for antibody design[18,19]. By training on natural antibody repertoires, Shin et al.[19] were able to design antibody libraries that display good physical properties and are enriched for binders. In the future, our approach can be combined with these by using a pre-trained generative language model to design the initial mutagenesis library used for training our supervised learning approach. Our initial analysis indicates that this approach is likely to increase the success rate of the initial library by several fold. Furthermore, pre-trained models could also condition on features of the target epitope to design target-specific initial libraries that are then fine-tuned with our framework.

We demonstrate the ability to rapidly design large libraries of potently binding scFvs, but our framework also extends to other domains of protein engineering where large scale functional mutagenesis screens are being applied. Our framework is neither scFv nor binding-specific and, therefore, can be applied to engineer other proteins for other functional properties. We expect machine learning approaches like ours, combined with high throughput mutagenesis screens, will soon become the standard in protein engineering.

## Methods
### Training data for language models
We used sequences from Pfam[27] and Observed Antibody Space (OAS)[28] databases to train four separate language models (i.e., a protein language model, an antibody heavy chain model, an antibody light chain model and a paired heavy-light chain model). The Pfam is a database of curated protein families containing raw sequences of amino acids for individual protein domains. We use the same data splits as provided in TAPE[8]. The train, validation and test splits contain 32,593,668, 1,715,454 and 44,311 sequences, respectively. The full OAS database contains immune repertoires from over 75 studies containing a diverse set of immune states. We curated only studies with naïve human

subjects and removed redundant sequences across the studies. This results in 37 studies containing 270,171,931 heavy chain sequences, 9 studies containing 70,838,791 light chain sequences, and 3 studies containing 33,881 heavy-light sequence pairs. The train, validation and test sequences are split based on studies. Given that there are limited heavy-light sequence pairs in the OAS data, to train the paired heavy-light chain model, we used all the data from OAS heavy chains, OAS light chains and OAS heavy-light sequence pairs. For sequence pairs with missing heavy or light chain, we left the missing chain as an empty sequence. Supplementary Table 3 summarizes the number of sequences in train, validation and test data for the four language model training datasets.

### Training BERT language models
We used the BERT masked language model[23] to encode protein/antibody sequences (Supplementary Fig. 12). The BERT model estimates the probability of an amino acid sequence p($\mathbf{x}$) by considering the probability distribution over each amino acid at each position conditioned on all other amino acids in the sequence, that is,

$$p(\mathbf{x}) = \prod_{i=1}^{i=L} p(x_i | x_1 \dots x_{i-1}, x_{i+1} \dots x_L) \tag{1}$$

where $x_i$ represents the $i^{th}$ amino acid in the sequence of length L. We pretrained four separate BERT language models, i.e., a protein language model, an antibody heavy chain model, an antibody light chain model and a paired heavy-light chain model, using the Pfam data and OAS data. Specifically, BERT masked language models were trained with 768 input embedding size, 24 hidden layers, 1024 hidden size, 4096 intermediate feed-forward size and 16 attention heads. All the other architecture details are fixed to their default values used in BERT[8,23] with Adam optimization[32]. We trained the language model to predict randomly masked amino acids in a single sequence or a sequence pair (Supplementary Fig. 12). For training the protein language model, antibody heavy chain model and antibody light chain model, the input is a single sequence of amino acids. For training the paired heavy-light chain model, the input is a concatenation of heavy and light sequences separated by a special token. Token type IDs are set to 0 for the 'CLS' token, 1 for the heavy chain amino acids and 2 for the light chain amino acids to identify two types of chains. Position IDs are set to be the integer position of the amino acid within its respective chain. The Pfam language model was initialized randomly. All other language models were initialized with the pre-trained Pfam model. For all models, the learning rate is set to $10^{-5}$, batch size is 1024 and the warm-up step is 10,000. One training epoch is defined as one full iteration over all the sequences in the training data. All models were trained until convergence of the cross-entropy loss value (which is evaluated on the validation data after every epoch), or until the maximum number of epochs, 10, was reached. All models were implemented in PyTorch[33] and trained on NVIDIA Volta V100 GPUs using a distributed compute architecture.

The standard average perplexity score is used to evaluate the language model performance on the hold-out test data. The perplexity measures how well the trained language models are at predicting the masked tokens. Lower values indicate better performance. The average perplexities of the 4 language models on the respective test data are 13.15 for the Pfam model, 1.56 for the heavy-chain model, 1.43 for the light-chain model and 1.16 for the paired model. When evaluated on the OAS light-chain test data, the average perplexities of the 4 language models are 7.47, 16.40, 1.43 and 1.42, respectively. When evaluated on the OAS heavy-chain test data, the average perplexities of the 4 language models are 12.20, 15.30, 1.56 and 1.56, respectively.

### Training sequence-to-affinity models via transfer learning
To prepare the training data[25], we randomly split the sequences in the initial Ab-14-H variant library and Ab-14-L variant library into train/

validation/test sets with 0.8/0.1/0.1 split. Since the experimental assay on the initial random scFv library was conducted in triplicate[25] (each scFv sequence has 3 measurements), the average value of all measurements corresponding to the same scFv is used. An assay with an empty measured binding affinity indicates that it is beyond the limit of detection and is deemed a poor binder. We considered two options for how missing values are treated: dropping the assay with missing value or imputing it with the median value of all assays of the same candidate chain.

We trained separate target-specific sequence-to-affinity models for Ab-14-H variants and Ab-14-L variants. We used model fine-tuning as a way to transfer knowledge learned from pre-trained language models to predicting sequence affinities. We investigated two approaches, which in addition to affinity prediction, provide estimates of prediction uncertainties: an ensemble method and Gaussian Process (GP). Both approaches use learned knowledge from pretrained language models and provide meaningful sequence-to-affinity models from which one can design a diverse antibody library.

The ensemble model consists of 16 different trained regression models that were fine-tuned from the 4 pretrained language models with two different regression loss functions and two different data preprocessing steps (Supplementary Table 9). The two loss functions used were the mean squared error (MSE) and the mean absolute error (MAE) between the predicted affinities and measured affinities. For the data preprocessing step, we used two options for treating missing values: dropping the assay with missing value or imputing it with the median value. To train a regression model, we fine-tuned the pretrained BERT language model (initially trained on massive sequence data without affinity measurements) by adding a linear regression decision head to the BERT model and continuing to train it on a smaller set of scFv sequences with experimental binding measurements. The outputs of the ensemble model are the mean and the standard deviation of the outputs of the 16 regression models.

While the ensemble method is known to enhance predictive performance, GP is another powerful technique used for quantifying uncertainties. For the GP model, we used the pretrained heavy-chain language model to train the GP model for the heavy chain sequence-to-affinity model and the pretrained light-chain language model for the light chain sequence-to-affinity GP model. Sequences were represented by first concatenating the learned vector representations of each amino acid from the pretrained language model, and then performing principal component analysis (PCA) to reduce the vector dimension to 1024. The GP model was trained on these reduced vector representations. Assays with missing values were imputed with the median value in the data preprocessing step. The trained GP model outputs a mean and a standard deviation of the binding affinity prediction.

## ML-extrapolated fitness functions

To generate a high affinity scFv library in silico, we used a Bayesian-based acquisition function extrapolated from the sequence-to-affinity model to construct the scFv fitness landscape. In contrast to non-Bayesian settings where the sequence is mapped directly to estimated affinity, the fitness function is defined to be a mapping from the entire scFv sequence to a posterior probability,

$$f(\mathbf{x}) = p(\text{aff}(\mathbf{x}) < \sigma | \mathbf{x}), \tag{2}$$

that the estimated binding affinity $\text{aff}(\mathbf{x})$ of the sequence $\mathbf{x}$ is better than the threshold $\sigma$. The threshold was set to the averaged assayed value of Ab-14 in the training data. Assuming a Gaussian distribution, $f(\mathbf{x})$ can be computed using the mean and standard deviation of the prediction from the trained sequence-to-affinity model. For each scFv chain (Ab-14-H and Ab-14-L), we computed two fitness functions, extrapolated from the ensemble model and GP model, respectively.

The proposed fitness function captures the model uncertainty during the optimization and enables us to estimate the performance of our antibody designs prior to experimental testing.

## Optimization strategies via sampling

The goal is to sample scFv sequences with the highest extrapolated fitness value $f(\mathbf{x})$. The optimization was performed using 3 different sampling algorithms: a greedy algorithm called hill climb (HC)[34], an evolutionary algorithm called genetic algorithm (GA)[35] and Gibbs sampling[36]. We initialized the HC and GA sampling processes using the 10 strongest binders (seed sequences) from the supervised training data and the Gibbs sampling using the strongest binders from the training data.

For the hill climb algorithm, we initialized the optimization by randomly mutating a seed sequence with an expected number of $k = 2$ mutations. At each step, the algorithm performs a local search around the current sequence and samples the next sequence that has the highest fitness value. The search continues until it can no longer find a sequence that has a better fitness value than the current sequence. We defined the local search space to be the 1000 mutants of the current sequence, consisting of all the $k = 1$ mutations and random $k = 2$ mutations. The greedy-based hill climb was run 100 times with random restart around a random seed sequence.

The genetic algorithm (GA) is an evolution-based search heuristic, where the fittest individuals are selected to produce offspring of the next generation. We initialized the population with a random seed sequence from the top 10 binders. Parents were chosen from the current population based on the Wright-Fisher model of evolution[37] where members of the current population become parents with a probability exponential to their fitness values, that is, $p(\mathbf{x}) \sim \exp(f(\mathbf{x})/\beta)$. Sequences with high fitness have more chances to pass their genes to the next generation. A single-point crossover was performed on two parent sequences randomly selected from the parent population, and followed by randomly mutating individual child sequences with an expected $k = 1$ mutation. The algorithm was terminated when it no longer produced new sequences (the population converged). The algorithm was run 100 times; each was initialized from a random seed sequence. The parameter $\beta$ was set to be 0.2 for the ensemble-based fitness function and 0.5 for the GP-based fitness function. Note that the selection of parameter value $\beta$ directly affects the diversity of generated sequence designs. Depending on the design needs, one can tune this parameter to adjust the overall library diversity. Due to limited understanding of the extrapolation power of ML models at the time of sequence design, the $\beta$ parameter was manually selected around its default value used in FLEXS[38]. Future work in applying the proposed in silico performance metric (see the Result section) to explore the tradeoffs between library diversity and percent of success would facilitate the selection of the β parameter.

Gibbs sampling is a Markov Chain Monte Carlo (MCMC) algorithm that samples a sequence according to some joint distribution by generating random variates from each of the full conditional distributions. We initialized the algorithm from the top seed sequence (the sequence with the strongest binding affinity in the training data). At each step, we randomly selected a position i in the sequence, sampled a mutant $\hat{x}_i$ at the selected position with a conditional probability,

$$p(x_i | x_1, \dots x_{i-1}, x_{i+1}, \dots x_L), \tag{3}$$

and updated the sequence by replacing the $i^{\text{th}}$ token with the sampled token $\hat{x}_i$. The conditional probability was defined to be exponential to the fitness values, that is,

$$p(x_i | x_1, \dots, x_{i-1}, x_{i+1}, \dots, x_L) \sim \exp(\gamma * f(\mathbf{x})). \tag{4}$$

The Gibbs sampling was run once with 30,000 iterations. The value $\gamma$ was set to be 18 for the Ab-14-H ensemble-based fitness function, and 20 for both the Ab-14-L ensemble- and GP-based fitness function. Multiple $\gamma$ values were used to sample the Ab-14-H GP-based fitness function. This is due to the limited number of sequences that can be sampled at any specific $\gamma$ value for the given fitness function. To ensure that enough sequences can be sampled, we used $\gamma = 10, 3, 2$, and ran the Gibbs algorithm three times to sample a sufficient number of sequences.

## ML-optimized ScFv libraries

For each scFv chain (Ab-14-H variants and Ab-14-L variants), we constructed two fitness functions extrapolated from the ensemble and GP model, respectively. For each fitness function, we performed optimization using three sampling strategies. This resulted in 6 libraries per chain: 3 libraries from optimizing the ensemble-based fitness function (namely, En-HC, En-GA and En-Gibbs), and 3 libraries from optimizing the GP-based fitness function (namely, GP-HC, GP-GA, GP-Gibbs). We then rank-ordered the generated sequences based on their fitness score per library and selected the top 6000 sequences per library for experimental validation. Supplementary Figs. 3 and 4 show the distribution of the designed sequences with respect to various mutational distances to demonstrate the library diversity: (1) mutational distance to the candidate scFv Ab-14, and (2) pairwise mutational distance in a library. The first distance metric measures the number of mutations the designed antibodies are from Ab-14. The second distance metric measures the intra-library diversity.

## Evolution directed libraries

We built two baseline libraries based on conventional directed evolution strategies: random mutations and the PSSM-based method. The random mutation library was constructed by randomly mutating amino acid tokens from the seed sequences in the training data with a $k = 2$ average number of mutations. Using this method, 2097 Ab-14-H heavy-chain variants and 477 Ab-14-L light-chain variants were generated for experimental testing.

For the PSSM-based library, we used sequences in the training data with measured affinities that are as good or better than the candidate scFv Ab-14. We fitted the PSSM by counting the occurrence of each amino acid at each position in the CDRs with a small pseudocount. The fitted PSSM is a matrix of probability scores for each amino acid at each position, representing the statistical patterns of the training sequences that are better than Ab-14. We then drew samples to generate designs based on the fitted PSSM. Contrary to the random mutation approach, the PSSM-based approach is not restricted to a pre-defined mutational distance and could generate sequences that are potentially far from the candidate antibody if the computed PSSM allows. The PSSM method resulted in 7748 Ab-14-H heavy-chain variant designs and 8257 Ab-14-L light-chain variant designs that were sent for experimental testing. Supplementary Fig. 5c, f shows the distribution of the generated sequences with respect to the mutational distances.

## Experimental validation of designed sequences

We used an engineered yeast mating assay to empirically measure the relative binding strength of our ML-designed sequences. Yeast peptone dextrose (YPD), yeast peptone galactose (YPG), and synthetic drop out (SDO) media supplemented with 80 mg/mL adenine were made according to standard protocols. Suppliers used for our yeast media are as follows: Bacto Yeast Extract (Life Technologies), Bacto Tryptone (Fisher BioReagents), Dextrose (Fisher Chemical), Galactose (Sigma-Aldrich), Adenine (ACROS Organics), Yeast Nitrogen Base w/o Amino Acids (Thermo Scientific), SC-His-Leu-Lys-Trp-Ura Powder (Sunrise Science Products), Yeast Synthetic Drop-out Medium Supplements (Sigma-Aldrich), L-Histidine (Fisher BioReagents),

L-Tryptophan (Fisher BioReagents), L-Leucine (Fisher BioReagents), Uracil (ACROS Organics), and Bacto Agar (Fisher BioReagents).

AlphaSeq compatible plasmids encoding yeast surface display cassettes were constructed by Twist Bioscience and resuspended at 100 ng/μL in molecular grade water (Corning). 100 ng of plasmid was digested with PmeI enzyme (NEB) for 1 hr at 37 °C to linearize, leaving chromosomal homology for integration into the ARS314 locus at both the 5' and 3' ends[39]. Yeast transformations were performed with Frozen-EZ Yeast Transformation Kit II (Zymo Research) according to manufactures instructions. Yeast were plated on SDO-Trp plates and grown at 30 °C for 2-3 days. Successful transformants were struck out onto YPAD plates and grown overnight at 30 °C.

To validate protein expression, yeast were inoculated in YPAD and grown overnight at 30 °C. Yeast were labelled with FITC-anti-C-myc antibody (Immunology Consultants Laboratory, Inc.) in PBS (Gibco) + 0.2% BSA (Thermo Fisher Scientific) for 30 minutes at RT. Yeast were pelleted and resuspended in PBS + 0.2% BSA and read on a LSRII cytometer.

To construct the DNA library, a 300 bp oligonucleotide pool synthesized by Twist Bioscience was resuspended at 20 ng/μL in molecular grade water (Corning). Libraries were PCR amplified from the oligonucleotide pool using KAPA DNA polymerase (Roche). The oligonucleotide amplification fragment was inserted into the seed scFv backbone using Gibson isothermal assembly (NEB), as well as a second DNA fragment containing a randomized DNA barcode. The assembled barcoded antibody DNA library was PCR amplified. Fragments were run on a 0.8% agarose gel and extracted using Monarch Gel Purification kit (NEB).

For the yeast library transformation, MATa AlphaSeq yeast were grown for 16 hours in YPAG media to induce SceI expression[39]. All spin steps were performed at 3000 RPM for 5 minutes. Yeast were spun down and washed once in 50 mL 1 M Sorbitol (Teknova) + 1 mM CaCl$_2$ (Sigma-Aldrich) solution. Washed yeast were resuspended in a solution of 0.1 M LiOAc (ACROS Organics)/1 mM DTT (Roche) and incubated shaking at 30 °C for 30 minutes. After 30 minutes, yeast were spun down and washed once in 50 mL 1 M Sorbitol + 1 mM CaCl$_2$ solution. Yeast were resuspended to a final volume of 400 μL in 1 M Sorbitol + 1 mM CaCl$_2$ solution and incubated with DNA for at least 5 minutes on ice. Yeast were electroporated at 2.5 kV and 25 uF (BioRad). Immediately following electroporation, yeast were resuspended in 5 mL of 1:1 solution of 1 M Sorbitol:YPAD and incubated shaking at 30 °C for 30 minutes. Recovered yeast cells were spun down and resuspended in 50 mL of SDO-Trp media and transferred to a 250 mL baffled flask. 20 μL of resuspended cells were plated on SDO-Trp to determine transformation efficiency. Both the flask and plate were incubated at 30 °C for 2-3 days. After 2-3 days, transformation efficiency was determined by counting colonies on the SDO-Trp plate.

For nanopore barcode mapping, genomic DNA from yeast libraries was extracted using Yeast DNA Extraction Kit (Thermo Fisher Scientific) following the manufacturer's instructions. A single round of qPCR was performed to amplify a fragment pool from the genomic DNA containing the gene through the associated DNA barcode. qPCR was terminated before saturation to minimize PCR bias, generally between 15-20 cycles. The final amplified fragment was concentrated with KAPA beads (Roche), quantified with a Quantus (Promega), prepped with an SQK-LSK-110 ligation kit (Oxford Nanopore) and sequenced with a Minion R10 flow cell (Oxford Nanopore) following the manufacturer's instructions. Each sequencing read was aligned to the set of expected antibody sequences from the in silico antibody library using minimap2[40] to determine the mapping between DNA barcodes and antibody sequence; only DNA barcodes with at least 2 reads observed were considered, and each DNA barcode was matched to the most common minimap2 antibody match among its constituent reads.

Library-on-library AlphaSeq assays were performed. Two mL of saturated MATa and MATalpha library were combined in 800 mL of YPAD media and incubated at 30 °C in a shaking incubator. Six technical replicates were performed. After 16 hr, 100 mL of yeast culture was washed once in 50 mL of sterile molecular grade water (Corning) and transferred to 600 mL of SDO-lys-leu with 100 nM ß-estradiol (Sigma-Aldrich) for 24 hr. After 24 hr, 100 mL of yeast was transferred to fresh SDO-lys-leu with 100 nM ß-estradiol for an additional 24 hr. In addition to the antibody libraries described above, control yeast strains comprising a small network of BCL2-family proteins[39] were included in each experiment to act as a set of standards for which BLI-derived interaction affinities were known a priori.

To prepare the library for next-generation sequencing, genomic DNA was extracted using Yeast DNA Extraction Kit (Thermo Fisher Scientific) following manufacturer's instructions. qPCR was performed to amplify a fragment pool from the genomic DNA and to add standard Illumina sequencing adaptors and assay specific index barcodes. qPCR was terminated before saturation to minimize PCR bias, generally between 23-27 cycles. The final amplified fragment was concentrated with KAPA beads (Roche), quantified with a Quantus (Promega), and sequenced with a NextSeq 500 sequencer (Illumina).

Sequencing data were analyzed to identify the MATa and MATalpha barcode pairs present among diploid yeast. The observed number of sequencing reads for each MATa/MATalpha combination were normalized according to frequency among haploid yeast to account for uneven distribution of the input populations. Each aα pair was then assigned a score representing the ratio of observed sequencing reads to expected sequencing reads assuming random mating. A linear regression was performed comparing these normalized sequencing scores to known affinities for the control yeast strains and this regression was utilized to assign estimated affinities to all other aα pairs for each mating replicate.

Supplementary Tables 4 and 5 summarize the number and percentage of sequences present in the experimental data for Ab-14-H and Ab-14-L designs, respectively. All generated data with experimental affinity measurements are made publicly available for research use[26]. To use the experimentally collected affinity data for evaluating the performance of designed scFv sequences, we only consider designs that are present in the experimental data. For sequences that are present in the affinity data and have at least three out of six empirical affinity values, the values are averaged and used as ground-truth measured affinities. Sequences with two or fewer empirical measurements are considered poor binders, and are included in the performance evaluation as un-successful designs.

### T-SNE Embedding

T-Distributed Stochastic Neighbor Embedding (t-SNE)[41] is used to visualize high-dimensional scFv sequences while approximately preserving the edit distance between sequences. Specifically, we first encode all scFv sequences using a one-hot encoder; for any pair of one-hot encoded scFv sequences, the L1-norm between them equals the edit distance. Then we apply the t-SNE dimensionality reduction to project one-hot encoded sequences into a 2-D space as shown in Fig. 3. Python scikit-learn package[42] was used to perform t-SNE with the L1-norm and PCA initialization[43]. For Ab-14-H variants, the perplexity and learning rate are set to be 500 and 200, respectively. For Ab-14-L variants, the perplexity and learning rate are set to be 500 and 500, respectively.

### Biophysical property calculation, statistical analysis of libraries

For the biophysical property analysis of designed libraries, we computed isoelectric points and hydrophobicity, which are physicochemical descriptors known to influence the solution behavior of antibodies. These properties were calculated based on the sequences of the heavy and light chain variants in each library using BioPython[30].

Specifically, for the heavy chain, we concatenated each heavy-chain design with the fixed light-chain sequence; for the light chain, we concatenated the fixed heavy-chain sequence with each light-chain design. Isoelectric points were calculated using pK values[44-46]. Hydrophobicity was calculated using the Kyte & Doolittle index[47]. The hydrophobicity score of each amino acid was averaged over the sequence of each variant to give an overall hydrophobicity score for each sequence. Supplementary Fig. 11 shows the distribution of isoelectric and hydrophobicity.

### Statistics and reproducibility

All statistical calculations were performed within the computation environment Python (v3.8). No statistical method was used to determine the training data size and the size of designed sequences for each library. The training data size was chosen as the maximum number of sequences our budget for experimental measurements allowed for, while being statistically sufficient for analysis and machine learning models. The number of designed sequences for each library was chosen to be 6000, which is statistically sufficient for analysis and method comparison. All designed sequences and controls were tested. Sequences that were unsuccessfully mapped in the haploid step (thus no binding measurement is available) were excluded from the analysis. The random library generation was randomized at k =1,2 or 3. The PSSM-based sequences were randomly sampled based on the PSSM statistics. The empirical validation was conducted blindly by individuals lacking knowledge of which method generated a given design. Language model-based scFv feature representations, sequence-to-affinity models and fitness landscapes were reproducible.

### Reporting summary

Further information on research design is available in the Nature Portfolio Reporting Summary linked to this article.

## Data availability

All raw data, including both the training data and designed sequences, generated in this study have been deposited to Zenodo under https://doi.org/10.5281/zenodo.7783546. Both datasets are under Creative Commons Attribution-NonCommercial-ShareAlike 4.0 International Public License ("CC BY-NC-SA 4.0"). Note: We do not recommend combining the training data set and the set of designed measurements into a single dataset. Even though both are from the same measurement system, the dynamic range is different between the two sets (as expected) potentially resulting in disagreement between the reported values for a sequence that exists in both sets. Source data are provided with this paper.

## Code availability

Code[48] is available on github: https://github.com/AIforGreatGood/biotransfer and on Zenodo under https://doi.org/10.5281/zenodo.7927152 for academic and/or non-profit internal research purposes.

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

## Acknowledgements

We would like to thank the scientists at A-Alpha Bio for assistance in generation of data; Eric Schwoebel, Joshua Dettman, and Tim Lu for thoughtful discussion of the research program and approach; Jack McGowan and Irene Stapleford for graphic design support; and the many other colleagues at MIT LL who have supported this project. This research was funded by the Under Secretary of Defense for Research

and Engineering under Air Force Contract No. FA8702-15-D-0001. Any opinions, findings, conclusions or recommendations expressed in this work are those of the author(s) and do not necessarily reflect the views of the Under Secretary of Defense for Research and Engineering.

## Author contributions

R.C., T.B., and M.W., conceived the project; R.C. and M.W. designed the data collection process; L.S. performed in silico mutagenesis; L.L., R.C., and T.B. designed model architectures; L.L., E.G., J.S., and T.B. performed algorithm development; L.L., E.G., J.S., and L.S. trained models; L.L., E.G., and L.S. evaluated models; L.L., E.G., J.S., T.B., and M.W. performed data analysis; E.E. and R.L. generated empirical data; L.L., R.J., E.E., R.C., T.B., and M.W. wrote and edited the manuscript.

## Competing interests

T.B. is the co-founder and CEO of NE47 Bio Inc., a company that provides machine learning services and software for protein engineering. R.L. is a co-founder and current employee of AAlpha Bio, Inc. (A-Alpha Bio) and owns stock/stock options of A-Alpha Bio. E.E. is an employee of A-Alpha Bio and owns stock/stock options of A-Alpha Bio. The remaining authors, L.L., E.G., J.S., L.S., R.J., R.C. and M.W. declare no financial and non-financial competing interests. Massachusetts Institute of Technology has filed a provisional patent application (63/373682) on the end-to-end method for generating scFv libraries; inventors include L.L., E.G., J.S., L.S., R.J., R.C., T.B. and M.W. A-Alpha Bio has a patent (US10988759B2) relating to certain research described in this article.
