## [Peer Review File · Nature Communications]

Machine Learning Optimization of Candidate Antibody Yields Highly Diverse Sub-nanomolar Affinity Antibody LibrariesReviewer #1 (Remarks to the Author):

The authors present natural language processing neural networks applied to the task of antibody fragment optimization (only the variable region) and perform experimental validation through yeast display of these fragments. They characterize the binding affinities of the resulting antibody fragments and demonstrate high affinities (sub-nanomolar) (Table 1, Fig 2A and 2D) and high success rates (Fig 2B and 2E). The authors convincingly demonstrate the superiority of their method to traditional computational methods relying on evolutionary data (PSSMs) (Table 1, Fig 1C). Additionally, they demonstrate the resulting mutants are diverse (Fig 2C and Fig 2F, Fig 3), decreasing the likelihood of total failure in further experimental testing provided multiple constructs are proposed and used. In terms of neural network architectures, the authors use the well-known BERT architectures use the attention mechanism. This work appears novel, addresses an important biological problem with applications in the lab and in the clinic, and the experimental validation is thorough and convincing, so I highly recommend this manuscript for publication.

Comments:

1. In Figs 1A and 1C, what does the azimuthal (angular) position represent? The radial dimension is labeled as mutational distance, but I don't think it is explained what the angular component is showing.
2. Please consider formatting equations on separate lines to improve readability
3. Please consider including an image of the specific architecture (or at least a generic schematic) you used. It is easier for the curious reader to interpret such a labeled diagram than to read a laundry list of hyperparameters
4. Please consider citing some of the foundational papers for machine learning, including (1) frameworks used (TensorFlow, PyTorch, JAX, etc.) and (2) residual connection layers (3) dropout (4) the Adam optimization algorithm
5. DOI is not listed for data availability
6. GitHub link is not listed in code availability

Reviewer #2 (Remarks to the Author):

Li et al here show an experimental-computational approach for antibody optimization using a library approach.

The paper is very concisely written and provides an in depth characterization of a large number of parameters tested.

I mostly have clarifying questions:

- it's not very clear what the antigen is. Can the authors elucidate on this?
- The authors give the impression that the findings are antigen-independent. but have they shown this?
- for Bayesian antibody optimization, the authors might also want to cite: [https://www.cell.com/cell-reports-methods/pdf/S2667-2375\(22\)00276-4.pdf](https://www.cell.com/cell-reports-methods/pdf/S2667-2375(22)00276-4.pdf)
- to what extent is your approach dependent on a language model? Would simpler encodings have worked as well?
- What do you mean by "end-to-end": can you define this term?
- How sequence and biophysical property-diverse are the ML-generated sequences compared to the training data?
- How much cost and time does it save compared to traditional approaches?
- The authors use the word target-specific. Would this approach also enable epitope-specific design – since, i guess, for now the sequences generated have a number of epitopes?

Reviewer #3 (Remarks to the Author):

In this manuscript, the authors describe an end-to-end Bayesian, language model-based method to design large and diverse libraries of high-affinity scFvs. In addition to demonstrating the significant impact of machine learning models to generate scFvs, the proposed approach may be of relevance to surpass major time-consuming gaps in antibody development. Also, beyond scFvs, there is potential to consider high-scored variable heavy chain (VH) and variable light chain (VL) sequence variants to obtain monoclonal antibodies in different molecular formats.

Comments:

1. The antibody schematic representation in Fig. 1 may be revised.

(a) Based on the report by Engelhart et al (ref. 23) and the described in the manuscript, the candidate antibody identification started with a library of phages displaying Fab fragments, not full-length immunoglobulins, which is what is shown in Fig. 1A. Also, the considered target was a peptide (LCBiot-PDVDLGDISGINAS-OH), and the image should reflect this.

(b) Fig. 1D provides a scheme for the machine learning-driven scFv design process. Despite the images appearing along with the "sequences with binding affinity measurements" indication, affinity data were not obtained from full-length immunoglobulins. It is recommended to adapt the illustrations to the performed steps.

2. It is recommended to present scFvs, the explored artificial proteins, as just "scFv" instead of "scFv antibody" (Fig. 1 legend; page 2). Also, to refer to the machine learning-designed library, "scFv libraries" is better than "antibody libraries".

3. The manuscript includes data on pI and hydrophobicity, physicochemical descriptors known to influence the solution behavior of antibodies. The pI values calculated for most of the considered Ab-14-H variants were in the 8.0-10.0 interval. The exception is in the ensemble-based method, in which most variants were described to have acidic pI, below 6.0. And curiously, the best affinity values of Ab-14-H variants were found in two of the ensemble-based libraries (En-HC and En-GA).

(a) The biophysical data in Supplementary Fig. 11 were weakly described. It is desirable an improved discussion on that, including the pI and affinity findings pointed out above.

(b) There has been described some association between the pI of Fv regions and the antibody colloidal stability. It was reported, for example, that aggregation-resistant VH domains tend to have acidic pI (Arbabi-Ghahroudi et al., 2009; Dudgeon et al., 2009; Nilvebrant et al. 2016), and that Fvs with high pI values show higher aggregation propensity at physiologic pH than those with low pI (Heads et al., 2019). It is known that scFvs are artificial constructs that, under certain conditions, can have low solubility and be prone to aggregation, which may interfere with their binding activity. Considering that, the detected differences in pI values with the ensemble-based method, and wondering whether scFv stability had any role in the binding results:

- (i) Any data or control about the proper folded percentage/yield of the scFvs displayed on yeast surface? The high-throughput yeast expression AlphaSeq system was used for both generating sequence-to-affinity model training and for experimental validation of designed variant affinity. This information might be also useful to exclude an eventual association of less stable scFvs from low pI value. An improved methodology description may be of help here.

- (ii) The provided AlphaSeq methodology was too short. More details of the method are desirable.

(c) The described machine learning approach may "lead to the design of diverse target-specific scFv libraries with therapeutically relevant binding affinities". For therapeutic antibodies, it would be interesting to include a brief discussion on the selection of libraries considering not only the affinity of the variants, but also the physicochemical properties (such as pI value) that have a potential in clearance and tissue retention.

4. About the mention of results in the manuscript:

(a) The reference to figures and tables in the text should be rechecked. Supplementary Tables 4, 5, 6, and 7 are mistakenly referred to in the "Online methods" section as Supplementary Tables 3, 4, 5, and 6, respectively. "The binding distribution of selected libraries" indicated on page 14 is found in Supplementary Fig. 6, not 4. Supplementary Fig. 1 was not referred to in the manuscript. Supplementary Tables 1 and 2 are also not listed in the text.

(b) In addition to not being mentioned in the manuscript, Tables 1 and 2 point out "three candidate antibodies", which initially contrasts with the manuscript message that "all heavy- and light-chain sequences in [the supervised training] data were generated by performing random k=1,2,3 mutations of the candidate antibody Ab-14" (page 4). It is required to improve data

presentation and description. Also, the too-short methodology provided to describe the experimental training data may have contributed to such concern.

(c) It is recommended to have the figures provided in the same order as they appear referred to in the text.

(d) It is advisable to change "...outperform all GP-generated libraries (59.4% - 84.2%)" (page 5) to "...outperform all GP-generated Ab-14H variant libraries (59.4% - 84.2%)".

5. About the methodology:

(a) Regarding the applied dimensionality reduction technique, although the last "Materials and methods" subsection lists uniform manifold approximation and projection (UMAP) in its title, the method for running this algorithm was not provided. But beyond that, it is not possible to know whether this technique was indeed used or not. Fig. 3 provides t-distributed stochastic neighbor embedding (t-SNE) projections, whose methodology was not described too. This should be verified.

(b) Despite the provided references (no. 23 and 27), the "Experimental Binding Measurements for Sequence-to-Affinity Model Training" method section should be improved.

6. Concerns about the use and presentation of references:

(a) Refs. 33 and 34 (page 4) are not suitable to refer to the Pfam data and the Observed Antibody Space (OAS) database, respectively.

(b) The report, by "Shin et al. [22]", describing the design of "antibody libraries that display good physical properties and are enriched for binders" is found in reference 19, not 22.

(c) It is recommended to include in ref. 35 the website (possibly <http://fields.scripps.edu/DTASelect/20010710-pl_Algorithm.pdf>) and the year of the last update (2003) of the referred document.

(d) Ref. 23 (Engelhart et al., 2022), provided in an "in press" format, should have its citation updated.

7. Figure legends should be improved to enhance the description of the related content.

(a) In Fig. 3 legend, there is no mention to what refer the red star (the best scFv variant in the PSSM library) and the red circles (the best variants for each other considered library) found in the two-dimensional scatterplots.

(b) For Supplementary Figs. 1, 6, and 10A: x-axis title ("log₁₀ Kd (log₁₀(nM))") should be simplified to something like "log₁₀ [Kd (nM)]" or "Kd (log₁₀ nM)".

(c) When describing measured values, it is desirable to emphasize whether the provided data come from predictive or empirical analyses.

8. Typo: "best design is 2818% better..." (page 2) instead of "best design is 28.8% better...".

Reviewer #4 (Remarks to the Author):

In the manuscript by Li et al, the authors develop a machine learning based method to design large and diverse libraries of single-chain variable fragments against a SARS-CoV-2 target peptide and show experimentally that the ML-generated libraries outperform both random and PSSM libraries. They also show that a significant number of new variants have affinities exceeding those of the candidate antibody Ab-14 fetched from a naive library of human antibodies using phage display. The authors used the BERT language model to map scFv sequences to predicted binding affinities in a probabilistic manner. The network was first pre-trained on sequences from Pfam and Observed Antibody Space databases on the masked token prediction task and then fine-tuned on the binding affinity prediction task using a dataset of 100k experimental binding affinities measured by the AlphaSeq assay for the same target-scFv system and published recently in a separate study. The trained models predict both binding affinities and respective uncertainties (standard deviations) which allowed the authors to devise and use several optimization strategies to design optimized scFv libraries, and the top sequences were then validated using the same AlphaSeq assay. The manuscript is well written overall and all the methods and results are clearly described for the most part. Here are a few suggestions on how the manuscript can be further improved:

- 1) Adding a sentence or two in the main text giving more details about the target may be helpful for the reader.
- 2) In addition to comparing performance of the newly generated libraries with the candidate Ab-14, it may be of interest to see how many good binders are present in the training set and what their affinities are (Fig 2). How much does the suggested method improve on top of the best binders in the training set?
- 3) It is curious that the ensemble model is much more robust at predicting affinities for high-order mutants compared to the GP model. What is the agreement between all the models in the ensemble? Is it possible to know how much using OAS on top of Pfam in pre-training helps for affinity predictions?
- 4) P.13, Methods: Did the authors do any fine-tuning of the beta parameter for sampling? If so, then some details should be provided.
- 5) The authors mention in the text (p.11) that AlphaSeq provides "predicted affinity" values. How are these values correlated with actual affinities? Has the assay been used before to assess protein-peptide interactions? Are there any differences in performance compared to assaying protein-protein interactions?
- 6) I was not able to find references to supplementary tables 1-3 and figs. 1,6 in the main text.

Response to reviewer comments:

We thank the reviewers for their time and thoughtful comments in reviewing the manuscript. The following are our responses to the reviewers:

Reviewer #1:

1. In Figs 1A and 1C, what does the azimuthal (angular) position represent? The radial dimension is labeled as mutational distance, but I don't think it is explained what the angular component is showing.

Author response: The circular plot (including the azimuthal position) is a conceptual description of the sequences used during training and those generated by ML. In this conceptual diagram, we show sequences at k mutations from the candidate sequence (the center of the diagram); each circle represents the sequence space with its associated mutation number. For training data, we generated $k=1, 2, 3$ mutations uniformly at random from the candidate sequence, therefore we illustrate in the diagram that sequences are placed uniformly at random along each mutation circle (Fig. 1A). In the case of ML-optimized sequences, we highlight the role of the optimization task in finding targeted areas in the space of potential mutations (Fig. 1C). Our results demonstrate that this is indeed the case - our ML approach ends up finding sequences with high binding properties and surprisingly at much higher mutational distance.

2. Please consider formatting equations on separate lines to improve readability

Author response: We have put equations on separate lines. We agree with the reviewer that this improves readability.

3. Please consider including an image of the specific architecture (or at least a generic schematic) you used. It is easier for the curious reader to interpret such a labeled diagram than to read a laundry list of hyperparameters

Author response: We thank the reviewer for the suggestion. We have added the BERT transformer architecture to the supplementary material (Supplementary Fig. 12).

4. Please consider citing some of the foundational papers for machine learning, including (1) frameworks used (TensorFlow, PyTorch, JAX, etc.) and (2) residual connection layers (3) dropout (4) the Adam optimization algorithm

Author response: We have added citations to relevant machine learning papers, e.g., PyTorch, Adam optimization and BERT which is the architecture our model is based on.

5. DOI is not listed for data availability

Author response: We have added the data DOI. We thank the reviewer for pointing this out. We also added a statement on the data availability for the designed sequences. We'll provide the DOI number once authors' institute completes the review on the data license.

6. GitHub link is not listed in code availability

Author response: We have added the Github link to code. Note that the link might not be public yet. We'll make it public once authors' institute completes the review of the code license and upon publication of

the work. The reviewers should have access to the zipped version of the entire code base with examples and demos.

Reviewer #2:

1. It's not very clear what the antigen is. Can the authors elucidate on this?

Author response: We have now added an additional sentence in the second paragraph of the Results section that now explains the target peptide is from the HR2 region of the spike protein.

2. The authors give the impression that the findings are antigen-independent. but have they shown this?

Author response: The process of training our ML models and generating scFv designs did not involve the sequence or other features of the antigen. The antigen is only used in experimental binding quantification to generate the affinity values used in the training data and perform empirical evaluation. Hence, the ML method is independent of the choice of the antigen and is generalizable to any antigen of interest.

3. For Bayesian antibody optimization, the authors might also want to cite: [https://www.cell.com/cell-reports-methods/pdf/S2667-2375\(22\)00276-4.pdf](https://www.cell.com/cell-reports-methods/pdf/S2667-2375(22)00276-4.pdf)

Author response: We have added citation to the Bayesian antibody optimization paper. We thank the reviewer for the suggestion.

4. To what extent is your approach dependent on a language model? Would simpler encodings have worked as well?

Author response: In a separate study (“Antibody Representation Learning for Drug Discovery.” arXiv, Oct. 05, 2022. doi: 10.48550/arXiv.2210.02881), we performed an extensive study on the role of different encoding methods in predicting binding affinity. We compared different encoding schemes, including a simpler encoder method such as the Position Specific Scoring Matrices (PSSM) encoder and various language models. We showed across a variety of affinity prediction methods (ridge regression, gaussian processes and a multilayer perceptron) that language models consistently outperform simpler encoding methods and that the more diverse the language model, the better the performance. We have added this detail in the Discussion section.

5. What do you mean by “end-to-end”: can you define this term?

Author response: The end-to-end terminology references the 3 components of our approach: the training data generation component (Fig. 1A), the ML-driven design component (Fig. 1B), and the empirical validation of designed scFvs component (Fig. 1C), all supporting the end goal of recommending new designs. Given a target antigen and a candidate antibody that binds weakly to the target (i.e., the inputs to our approach), the *training data generation* component samples the scFv space around the initial candidate and performs binding quantification. The *ML-driven design* component then takes the training data (sampled scFv sequences and corresponding binding measurements) and performs scFv optimization to find a library of strong binders. The last step, the *empirical validation* component validates the quality of the designed scFv library. The resulting scFv library with binding quantification provides a pool of potential candidates for development and is the output of our approach. We have added a clarification of this end-to-end process in the caption of Fig. 1.

6. How sequence and biophysical property-diverse are the ML-generated sequences compared to the training data?

Author response: We thank the reviewer for the suggestion. We have added the biophysical property comparison of ML-generated sequences and the training data to Fig. 11 in the supplementary materials. We observe that sequences generated by the ensemble method had more diverse biophysical properties, especially in the case of the isoelectric point.

When it comes to sequence diversity, we give a detailed analysis of the ML-generated sequence diversity when compared to training data (see page 5 and Fig. 2 and Fig. 3). We show that ML-generated sequences are highly diverse, sometimes as far as ~23 mutations away.

7. How much cost and time does it save compared to traditional approaches?

Author response: We agree with the reviewer that this is an important consideration for real-world implementation of our work. However, it is challenging to make a defensible calculation of the true savings that would be realized at a biotechnology company. That being said, we could attempt to make some rough comparisons between the ML approaches with the PSSM approach (“traditional approach”) using the data we have.

If the goal is to generate sequences that are improvements over the candidate, all ML heavy-chain libraries (except EN-Gibbs) generated 3-4 times more successful sequences than the PSSM library, meaning that the PSSM library would require at least 3-4 times more sequence designs to achieve the same number of successful sequences, but with significantly less sequence diversity. Because the AlphaSeq assay is a pooled approach, the financial cost of the difference is represented by the cost to synthesize additional oligos.

If the goal is to explore as much of the sequence space as possible, then the ensemble-based methods would result in tremendous time and cost savings. The average mutational distance of the heavy-chain ensemble libraries ranges from 7.9 to 15.6 while the PSSM library is 3.17 (Supplementary Fig. 3). One would expect at least 2-5 more rounds of PSSM design cycles to get to a mutational distance that is as far as ensemble libraries. For reference, a single design-build-test cycle is on the order of a few months. This difference is further exacerbated when considering that less than 25% of the empirically evaluated PSSM sequences are successful, compared to the >90% of success for En-GA and En-HC.

Another comparison could be between the experimental binding affinities of top scFvs. The PSSM approach resulted in a top scFv with an empirical affinity of 109.6 pM, while the En-HC method achieves nearly a 29-fold improvement over the PSSM. Additional rounds of the PSSM designs would be needed to reach a similar affinity, which would incur additional time and cost.

Future work in understanding these tradeoffs will be critical for wider-scale adoption of machine learning methods within the biotechnology industry. We ensured that our evaluation process included the PSSM (and some random) sequences so we could make the comparisons that are presented in the Results section and addressed in the first paragraph of the discussion section. We have added a discussion on the potential cost and time saving in the discussion section.

8. The authors use the word target-specific. Would this approach also enable epitope-specific design – since, I guess, for now the sequences generated have a number of epitopes?

Author response: We appreciate the reviewer asking this question as it further emphasizes their earlier request to clarify the peptide target that we used. This approach does allow for epitope specific design; we used a linear peptide from the HR2 region as the target instead of using the full spike protein. Because the target peptide is 14 amino acids in length, we consider this to be a single epitope. Additionally, in the

yeast surface display assay, we include negative control MAT α yeast that do not express the target peptide. This allows us to ensure scFvs are binding to the target and not to a different yeast surface protein.

Reviewer #3:

1. The antibody schematic representation in Fig. 1 may be revised.

(a) Based on the report by Engelhart et al (ref. 23) and the described in the manuscript, the candidate antibody identification started with a library of phages displaying Fab fragments, not full-length immunoglobulins, which is what is shown in Fig. 1A. Also, the considered target was a peptide (LCBiot-PDVDLGDISGINAS-OH), and the image should reflect this.

(b) Fig. 1D provides a scheme for the machine learning-driven scFv design process. Despite the images appearing along with the “sequences with binding affinity measurements” indication, affinity data were not obtained from full-length immunoglobulins. It is recommended to adapt the illustrations to the performed steps.

Author response: We thank the reviewer for pointing out the inconsistencies between Fig. 1 and our experimental methods. We have made the following changes to the illustration: (1) using Fab fragments in the depiction of phage display; (2) labeling the target with the actual peptide used; and (3) using scFvs instead of the full-length immunoglobulins in Fig. 1D.

2. It is recommended to present scFvs, the explored artificial proteins, as just “scFv” instead of “scFv antibody” (Fig. 1 legend; page 2). Also, to refer to the machine learning-designed library, “scFv libraries” is better than “antibody libraries”.

Author response: We thank the reviewer for the suggestion. We have changed all “scFv antibody” to “scFv”, and “antibody libraries” to “scFv libraries”

3. The manuscript includes data on pI and hydrophobicity, physicochemical descriptors known to influence the solution behavior of antibodies. The pI values calculated for most of the considered Ab-14-H variants were in the 8.0-10.0 interval. The exception is in the ensemble-based method, in which most variants were described to have acidic pI, below 6.0. And curiously, the best affinity values of Ab-14-H variants were found in two of the ensemble-based libraries (En-HC and En-GA).

(a) The biophysical data in Supplementary Fig. 11 were weakly described. It is desirable an improved discussion on that, including the pI and affinity findings pointed out above.

Author response: We thank the reviewer for pointing this out. We have added more details of biophysical property calculations in the Method section. In addition, we have also expanded the discussion as suggested by the reviewer in the caption of the supplementary figure.

(b) There has been described some association between the pI of Fv regions and the antibody colloidal stability. It was reported, for example, that aggregation-resistant VH domains tend to have acidic pI (Arbabi-Ghahroudi et al., 2009; Dudgeon et al., 2009; Nilvebrant et al. 2016), and that Fvs with high pI values show higher aggregation propensity at physiologic pH than those with low pI (Heads et al., 2019). It is known that scFvs are artificial constructs that, under certain conditions, can have low solubility and be prone to aggregation, which may interfere with their binding activity. Considering that, the detected differences in pI values with the ensemble-based method, and wondering whether scFv stability had any role in the binding results:

- (i) Any data or control about the proper folded percentage/yield of the scFvs displayed on yeast surface? The high-throughput yeast expression AlphaSeq system was used for both generating sequence-to-affinity model training and for experimental validation of designed variant affinity. This

information might be also useful to exclude an eventual association of less stable scFvs from low pI value. An improved methodology description may be of help here.

Author response: Unfortunately, we do not have access to this information. The scFvs were expressed in a pooled format, so we do not have stability or yield measurements for individual scFvs. As described in the updated Method section on experimental binding measurements for sequence-to-affinity model training, we have some negative control yeast that have no target on their surface. If the results were due to general aggregation, then we might have seen binding with negative controls. This suggests that our scFvs are binding target-specifically and are not reading out as high affinity due to general aggregation. In the reverse case, where aggregation would reduce the apparent affinity due to self-aggregation preventing interaction with the target, we can also rule this out as our library still contains high pI scFvs with strong binding affinities.

- (ii) The provided AlphaSeq methodology was too short. More details of the method are desirable.

Author response: We thank the reviewer for the suggestion and we now include an expanded AlphaSeq assay description in the manuscript.

(c) The described machine learning approach may “lead to the design of diverse target-specific scFv libraries with therapeutically relevant binding affinities”. For therapeutic antibodies, it would be interesting to include a brief discussion on the selection of libraries considering not only the affinity of the variants, but also the physicochemical properties (such as pI value) that have a potential in clearance and tissue retention.

Author response: This is a very interesting point, and we appreciate the reviewer bringing it up. We agree that this is an exciting possibility with machine learning-based design approaches like ours. In the future, biophysical properties, like pI, that are known to be associated with developability or physicochemical properties can be included in the library design criteria. As we showed here, our method was able to identify strong binders across a wide range of hydrophobicities and pIs which suggests that the library could have been designed to identify strong binders within specific pI or hydrophobicity ranges. We have now included a sentence on this in the discussion section.

4. About the mention of results in the manuscript:

(a) The reference to figures and tables in the text should be rechecked. Supplementary Tables 4, 5, 6, and 7 are mistakenly referred to in the “Online methods” section as Supplementary Tables 3, 4, 5, and 6, respectively. “The binding distribution of selected libraries” indicated on page 14 is found in Supplementary Fig. 6, not 4. Supplementary Fig. 1 was not referred to in the manuscript. Supplementary Tables 1 and 2 are also not listed in the text.

Author response: We thank the reviewer for taking the time to carefully read the manuscript and the supplementary material and point out these inconsistencies. We have checked and corrected all the references to figures and tables in the text. We have also added missing references in the main text.

(b) In addition to not being mentioned in the manuscript, Tables 1 and 2 point out “three candidate antibodies”, which initially contrasts with the manuscript message that “all heavy- and light-chain sequences in [the supervised training] data were generated by performing random k=1,2,3 mutations of the candidate antibody Ab-14” (page 4). It is required to improve data presentation and description. Also, the too-short methodology provided to describe the experimental training data may have contributed to such concern.

Author response: We appreciate the reviewer for identifying this discrepancy. Indeed, the training data does contain modified sequences of three candidate scFvs as suggested by Supplementary Tables 1 and 2. In this work, we only use data from one scFv, Ab-14. In the Results section, we have added the specifics of what exactly is included in the training data set for this work. We would also emphasize that we previously published the training data and the detailed process for generating this data, as a standalone entity to ensure we were able to clearly and extensively communicate its contents and considerations surrounding its use.

(c) It is recommended to have the figures provided in the same order as they appear referred to in the text.

Author response: We thank the reviewer for the suggestion. We have re-ordered all figures and tables so that their numbers are in the same order as they appear in the text.

(d) It is advisable to change "...outperform all GP-generated libraries (59.4% - 84.2%)" (page 5) to "...outperform all GP-generated Ab-14H variant libraries (59.4% - 84.2%)".

Author response: We thank the reviewer for the suggestion. We have changed the text accordingly.

5. About the methodology:

(a) Regarding the applied dimensionality reduction technique, although the last "Materials and methods" subsection lists uniform manifold approximation and projection (UMAP) in its title, the method for running this algorithm was not provided. But beyond that, it is not possible to know whether this technique was indeed used or not. Fig. 3 provides t-distributed stochastic neighbor embedding (t-SNE) projections, whose methodology was not described too. This should be verified.

Author response: We thank the reviewer for pointing this out. We have removed the UMAP reference and added the description for the visualization method used in this work (t-SNE) in the Methods section

(b) Despite the provided references (no. 23 and 27), the "Experimental Binding Measurements for Sequence-to-Affinity Model Training" method section should be improved.

Author response: We thank the reviewer for this feedback. We have added additional information to this section describing the yeast mating assay. Combined with the information added in response to this reviewer's comment 4(b), we hope this provides sufficient information for the reader to understand the training data sufficiently to understand the development of the machine learning framework.

6. Concerns about the use and presentation of references:

(a) Refs. 33 and 34 (page 4) are not suitable to refer to the Pfam data and the Observed Antibody Space (OAS) database, respectively.

(b) The report, by "Shin et al. [22]", describing the design of "antibody libraries that display good physical properties and are enriched for binders" is found in reference 19, not 22.

(c) It is recommended to include in ref. 35 the website (possibly <<http://fields.scripps.edu/DTASelect/20010710-pl Algorithm.pdf>>) and the year of the last update (2003) of the referred document.

(d) Ref. 23 (Engelhart et al., 2022), provided in an "in press" format, should have its citation updated.

Author response: We greatly appreciate the reviewer for pointing out the problems with references. We have made sure all citations are properly updated. We have also added the last update year to URL citation (Tabb, 2003), and removed the "in press" from Engelhart et al., 2022.

7. Figure legends should be improved to enhance the description of the related content.

(a) In Fig. 3 legend, there is no mention to what refer the red star (the best scFv variant in the PSSM library) and the red circles (the best variants for each other considered library) found in the two-dimensional scatterplots.

(b) For Supplementary Figs. 1, 6, and 10A: x-axis title (" $\log_{10} Kd (\log_{10}(nM))$ ") should be simplified to something like " $\log_{10} [Kd (nM)]$ " or " $Kd (\log_{10} nM)$ ".

(c) When describing measured values, it is desirable to emphasize whether the provided data come from predictive or empirical analyses.

Author response: We thank the reviewer for the suggestions. The legend to Fig. 3 is updated with a clear description of the red circles and red star. We have edited the x-axis to use " $Kd (\log_{10} nM)$ " for all relevant figures. We have also modified the legend to make sure that whenever "measured values" is used, it means empirical measurements.

8. Typo: "best design is 2818% better..." (page 2) instead of "best design is 28.8% better...".

Author response: We have corrected the inconsistency in the manuscript.

Reviewer #4:

1) Adding a sentence or two in the main text giving more details about the target may be helpful for the reader.

Author response: We thank Reviewer #4 for this suggestion which is aligned with that of Reviewer #2/question 1. We have added additional information to the main text describing the target (also see our response to Reviewer 2/question 1)

2) In addition to comparing performance of the newly generated libraries with the candidate Ab-14, it may be of interest to see how many good binders are present in the training set and what their affinities are (Fig 2). How much does the suggested method improve on top of the best binders in the training set?

Author response: We agree with the reviewer that it would be of interest to see the amount of improvement of ML designs from the training data. To this end, we have included a distribution comparison of the training data, PSSM designs and ML designs in Supplementary Fig. 12. We observe that ML designs are significantly better than the good binders in the training data. Notably, more than 25% of ensemble-based Ab-14-H variant designs have stronger binding affinities than the strongest binder in the training data, whereas only 0.9% of PSSM-based Ab-14-H variant designs outperform the strongest binder in the training data. We have included this observation in the "Experimental Validation of the Designed Sequences" section.

3) It is curious that the ensemble model is much more robust at predicting affinities for high-order mutants compared to the GP model. What is the agreement between all the models in the ensemble? Is it possible to know how much using OAS on top of Pfam in pre-training helps for affinity predictions?

Author response: The posterior predictions of a Gaussian process are weighted averages of the training data where the weights are based on the covariance and mean functions. It is known that the posterior predictions of a Gaussian process on out-of-distribution data have limited predictive power. Hence, its limited extrapolation power for high-order mutants, when the training data are of low-order mutants, is expected. On the other hand, Gaussian process can reliably separate in-distribution data from out-of-distribution data via its uncertainty quantification feature; hence it knows where its prediction limit is. As

a result, within its prediction limit, sequences generated from the Gaussian process model consistently exhibit a high success rate.

The agreement and disagreement between models in the ensemble model are captured by the standard deviation of the ensemble model in Supplementary Fig. 2C. Here we plotted the average standard deviation and the RMSE. There is a good alignment between the standard deviation and RMSE, indicating that models agree more (smaller standard deviation) when the predictions are more accurate and agree less (larger standard deviation) when the predictions are off, a result highlighting the consistent behavior of the ensemble method

In a separate study, (“Antibody Representation Learning for Drug Discovery.” arXiv, Oct. 05, 2022. doi: 10.48550/arXiv.2210.02881), we investigated the performance of Pfam and OSA language model in predicting binding affinity. The result suggests that the Pfam language model performs consistently better than the OSA language model. We postulate that while protein sequences in the Pfam dataset may exhibit differing underlying distributions, the diversity of the sequences in this dataset allows for learning additional higher level biological properties leading to improved affinity predictions. For the same reason, in this work, we chose to use a diverse set of language models to construct the ensemble. Unfortunately, we did not perform the experiment to investigate the value added of using OAS data on top of Pfam in pre-training, which would require a separate large-scale language model training. We have added an extended discussion on pretrained language models in the discussion section.

4) P.13, Methods: Did the authors do any fine-tuning of the beta parameter for sampling? If so, then some details should be provided.

Author response: We only adjusted the beta parameter when the resulting sequences appear to be too diverse or too similar because at the time of sequence generation, we had limited understanding of the extrapolation power of the ML models and the tradeoff between binding performance and sequence diversity. In fact, this is precisely the reason that we came up with the in silico performance metric, which would enable us to quantify the performance of an scFv library and perform beta parameter tuning to optimize the library performance. In this work, we have demonstrated that the proposed in silico performance metric can estimate the success rate of a scFv library. In the future work, one can apply this metric in the design process to fine-tune the beta parameter. In the manuscript, we have explicitly added the potential use of the proposed in silico performance metric for parameter tuning in the Result section, and also added more details on the beta parameter selection in the Method section.

5) The authors mention in the text (p.11) that AlphaSeq provides “predicted affinity” values. How are these values correlated with actual affinities? Has the assay been used before to assess protein-peptide interactions? Are there any differences in performance compared to assaying protein-protein interactions?

Author response: We thank the reviewer for bringing up this important point about our measurement system. For an in-depth look at the correlation, we would suggest looking at the Data Descriptor publication, “A dataset comprised of binding interactions for 104,972 antibodies against a SARS-CoV-2 peptide”. In this work we show that there is strong correlation ($R^2 = 0.85$) between our standards and BLI measurements. We do not have BLI measurements for any of our scFvs and thus cannot do this specific comparison. This is one reason why we often frame our results as fold-improvement as opposed to the absolute value.

As for protein-peptide interactions specifically, there is one BH3 peptide in the standard curve. Personal communication with scientists at A-Alpha Bio, LLC, confirms good performance with protein-peptide interactions in their system. It is also worth noting that yeast surface display systems have long been used

to measure protein-peptide interactions (example: Yang M, Wu Z, Fields S. Protein-peptide interactions analyzed with the yeast two-hybrid system. *Nucleic Acids Res.* 1995 Apr 11;23(7):1152-6. doi: 10.1093/nar/23.7.1152. PMID: 7739893; PMCID: PMC306824.) We would not expect meaningful differences in performance when comparing protein-peptide measurement to protein-protein measurement.

6) I was not able to find references to supplementary tables 1-3 and figs. 1,6 in the main text.

Author response: We thank the reviewer for pointing out the missing references. We have added the missing references to the main text.

Reviewer #1 (Remarks to the Author):

The authors have addressed all of the comments I raised during the first round of review. They have also improved the clarity of the manuscript in the discussion and figure captions. I would like to reaffirm that the manuscript is interesting and uses high-quality machine learning techniques in antibody fragment design. Finally, it appears that the other reviewers' comments have also been addressed.

Reviewer #2 (Remarks to the Author):

The authors have addressed all my comments.

Reviewer #3 (Remarks to the Author):

The authors answered the points raised by the reviewers and the revised manuscript brings improvements over the initial submission. An extended methodology, some enhanced discussion, and an important new comparison analysis between libraries were provided. General remaining concerns regarding the new content:

1. It was stated in the revised Supplementary Fig. 2 legend (lines 69-71) that "sequences are labeled as strong binders if the empirically measured affinities are smaller than the initial candidate sequence and weak binders if the empirically measured affinities are bigger than the candidate sequence" (Supplementary Fig. 2 legend; lines 69-71). However, the strong binders might be expected to have a lower K_d value, which corresponds to their higher (not smaller) measured affinity than that of the initial candidate sequence. It is advisable to verify that, and also provide an improved description of AUPR (area under the precision-recall curve).

2. In Fig. 1A and 1C histograms, it is advisable to change "candidate Fab" to "candidate scFv", in line with Fig. 1 legend and the mention on page 4 that "a phage library containing naïve human Fabs was used to identify candidate scFv".

3. It is recommended to consider rewriting the fragment "As indicated by our results (Table 1, Fig. 2 and 3), the conventional approach would need additional design- build-test cycles and synthesis of additional oligos, in order to achieve a comparable [...] percent of success as the ML approach" (lines 291-294). A discussion on the potential cost and time saving of the machine learning methods is of relevance. However, the conclusion pointed out in the segment cited above is not directly indicated by the results, despite that being stated. The manuscript would also benefit from further exploration of such subject.

4. Additional changes in the manuscript might be helpful to make the results easier to read, as the suggested below:

(a) It was added to the revised manuscript an empirical binding distribution comparison of the training data with the PSSM-derived and machine learning-derived scFv designs. It is worth considering describing these data in the "Results" section instead of the "Online methods".

(b) Minor suggestion: add a comma between "ML-optimized libraries" and "PSSM library" (lines 561-562) to increase clarity.

Reviewer #4 (Remarks to the Author):

I would like to thank the reviewers for carefully addressing all my comments. I have no further questions.

Response to reviewer comments:

We thank the reviewers for their time and thoughtful comments in reviewing the revised manuscript. The following are our responses to reviewers' comments.

Reviewer #3:

1. It was stated in the revised Supplementary Fig. 2 legend (lines 69-71) that "sequences are labeled as strong binders if the empirically measured affinities are smaller than the initial candidate sequence and weak binders if the empirically measured affinities are bigger than the candidate sequence" (Supplementary Fig. 2 legend; lines 69-71). However, the strong binders might be expected to have a lower Kd value, which corresponds to their higher (not smaller) measured affinity than that of the initial candidate sequence. It is advisable to verify that, and also provide an improved description of AUPR (area under the precision-recall curve).

Author response: We thank the reviewer for the suggestion. We have edited the caption to make it clear that strong binders have stronger measured affinities (lower Kd value) and weak binders have weaker measured affinities (higher Kd value). In addition, we added more detailed description on the AUPR and how it was computed.

2. In Fig. 1A and 1C histograms, it is advisable to change "candidate Fab" to "candidate scFv", in line with Fig. 1 legend and the mention on page 4 that "a phage library containing naïve human Fabs was used to identify candidate scFv".

Author response: We agree with the reviewer's suggestion. Fig. 1 is updated.

3. It is recommended to consider rewriting the fragment "As indicated by our results (Table 1, Fig. 2 and 3), the conventional approach would need additional design- build-test cycles and synthesis of additional oligos, in order to achieve a comparable [...] percent of success as the ML approach" (lines 291-294). A discussion on the potential cost and time saving of the machine learning methods is of relevance. However, the conclusion pointed out in the segment cited above is not directly indicated by the results, despite that being stated. The manuscript would also benefit from further exploration of such subject.

Author response: We thank the reviewer for the recommendation on further exploring the cost and time saving. As suggested by the reviewer, we have added additional comparison from our results to support the potential cost and time saving of using ML methods.

4. Additional changes in the manuscript might be helpful to make the results easier to read, as the suggested below:

(a) It was added to the revised manuscript an empirical binding distribution comparison of the training data with the PSSM-derived and machine learning-derived scFv designs. It is worth considering describing these data in the "Results" section instead of the "Online methods".

(b) Minor suggestion: add a comma between "ML-optimized libraries" and "PSSM library" (lines 561-562) to increase clarity.

Author response: We thank the reviewer for the suggestion. We have added the empirical binding distribution comparison between the training data and generated sequences to the Result section, and removed it from the Online methods.